# Highly Efficient MOF Catalyst Systems for CO₂ Conversion to Bis-Cyclic Carbonates as Building Blocks for NIPHUs (Non-Isocyanate Polyhydroxyurethanes) Synthesis

**Adolfo Benedito [1,\*], Eider Acarreta [1] and Enrique Giménez [2]**

[1] Instituto Tecnológico del Plástico (AIMPLAS), 46980 Valencia, Spain; eacarreta@aimplas.es
[2] Instituto de Tecnología de Materiales, Universitat Politècnica de València (UPV), 46022 Valencia, Spain; enrique.gimenez@mcm.upv.es
[\*] Correspondence: abenedito@aimplas.es

**Abstract:** The present paper describes a greener sustainable route toward the synthesis of NIPHUs. We report a highly efficient solvent-free process to produce [4,4′-bi(1,3-dioxolane)]-2,2′-dione (BDC), involving $CO_2$, as renewable feedstock, and bis-epoxide (1,3-butadiendiepoxide) using only metal–organic frameworks (MOFs) as catalysts and cetyltrimethyl-ammonium bromide (CTAB) as a co-catalyst. This synthetic procedure is evaluated in the context of reducing global emissions of waste $CO_2$ and converting $CO_2$ into useful chemical feedstocks. The reaction was carried out in a pressurized reactor at pressures of 30 bars and controlled temperatures of around 120–130 °C. This study examines how reaction parameters such as catalyst used, temperature, or reaction time can influence the molar mass, yield, or reactivity of BDC. High BDC reactivity is essential for producing high molar mass linear non-isocyanate polyhydroxyurethane (NIPHU) via melt-phase polyaddition with aliphatic diamines. The optimized Al-OH-fumarate catalyst system described in this paper exhibited a 78% GC-MS conversion for the desired cyclic carbonates, in the absence of a solvent and a 50 wt % chemically fixed $CO_2$. The cycloaddition reaction could also be carried out in the absence of CTAB, although lower cyclic carbonate yields were observed.

**Keywords:** MOFs; bis-cyclic carbonates; 1,3-butadiendiepoxide; $CO_2$; NIPHU; heterogeneous catalysis; carbon dioxide fixation

## 1. Introduction

Polyurethanes (PUs) are one of the most versatile classes of polymeric materials with excellent properties—namely, high durability, elasticity, water absorption, toughness, and abrasion resistance. Thus, they have widespread applications in many areas such as foams, footwear, coatings and paints, industrial machinery, adhesives, packaging, and medical devices [1]. The chemistry of polyurethanes (PUs) is well understood, and they have been utilized for many decades in daily life applications [2]. PU is the sixth most used synthetic polymer for industrial applications [3]. Recently PUs have been paid more attention by the chemical industry due to the industries' current dependency on fossil fuel resources, and the isocyanates presently used are toxic and dangerous to human health because they originate from phosgene.

Phosgene, $COCl_2$, is a volatile and extremely toxic gas at room temperature, with a boiling point of 8.3 °C [4]. When inhaled, it reacts with water in the lungs, forming hydrochloric acid and carbon dioxide. Exposure to these highly toxic gaseous compounds can pose serious health risks such as respiratory problems, eye and/or skin irritation, or even death [5]. Phosgene synthesis (Scheme 1) consists of reacting pure carbon monoxide and chlorine gas over an activated carbon bed, as a catalyst.

Conventionally, polyurethanes are synthesized via the polyaddition of petroleum-based isocyanates (derived from toxic phosgene and amines) with hydroxyl functional

compounds to generate urethane linkages [6], as shown in Scheme 2. As a result of the high reactivity and toxicity of isocyanates, they require special safety precautions and thus an adjusted manufacturing procedure [7].

**Scheme 1.** Synthesis of phosgene.

**Scheme 2.** Conventional synthesis of polyurethanes.

In the context of green chemistry and sustainability, it is highly desirable to use renewable resources and substitute toxic isocyanate intermediates [8]. During recent years, huge progress has been made regarding the development of non-isocyanate polyurethanes because of the increasing social demands for sustainability, low carbon footprint, bioeconomy, green chemistry, and human safety. A possible alternative would be the use of NIPHUs (non-isocyanate polyurethanes). NIPHUs are an interesting class of polymers based on the formation of 2-hydroxyurethane bonds via the reaction between cyclic carbonates and amines [9,10]. Some researchers [11,12] have proposed vegetable oils as a renewable feedstock that can be advantageously exploited as a reliable source to develop non-isocyanate polyhydroxyurethanes. However, the use of these vegetable oils would compete with food production [1]. Nowadays, the most attractive route reported for the formation on NIPHUs is through the ring-opening aminolysis of bis-cyclic carbonates with bis-amines, since cyclic carbonates are readily available via the chemical fixation of carbon dioxide, the notorious greenhouse gas [13]. Among the cyclic carbonates, five-membered cyclic carbonates (5CC) are considered to be the optimum type of CC because they react with primary amines and cyclic secondary amines.

The synthetic pathway of bis-cyclic carbonates to obtain NIPHUs is relatively easy and cheap since the typical process involves the reaction of an epoxide with $CO_2$ in the presence of a catalyst [10,14,15]. The use of $CO_2$, as a reactive, can be considered as a necessary step toward sustainable development for two reasons: firstly, it is possible to produce NIPHUs in an easier way, and secondly, it can contribute to green and sustainable technology, hence reducing emissions from greenhouse gasses. This novel approach removes phosgene as a

reagent and is 100% atom economical [16]. However, the slow reactivity of cyclic carbonates with amines unlike epoxy groups at lower temperatures is a major issue in the procedure. The lower reactivity results in a slower molecular build-up during polymerization [17,18]; hence, NIPHUs synthesis requires the addition of catalysts [19]. Even with this addition, the synthesis of NIPHUs needs both high temperatures and long reaction times of numerous hours [17,20,21]. Besse et al. concluded that the long polymerization time was due to side reactions that disrupt the precise stoichiometry [2,22,23]. It is an important goal in the synthesis of NIPHUs to enhance the polymerization rate of bis-cyclic carbonates without impacting stoichiometry in order to produce a high molar mass NIPHU within only a few minutes. Whelan et al. recognized that butadiendicarbonate (BDC) reacted with hexamethylendiamine (HMDA) in dimethyl sulfoxide solution at room temperature and without long reaction times to yield linear NIPHUs [24]. This pathway with BDC is more desirable than other pathways, with different intermediates, in the traditional synthesis of NIPHUs because BDC units are incorporated into the polyurethane backbone in the absence of protective groups; hence, it offers a great opportunity for producing NIPHU in an eco-friendly way. This review aims to shed light on recent progress than has been made in this direction.

The coupling reaction of $CO_2$ with bis-epoxides to synthesize bis-cyclic carbonates is an excellent pathway for $CO_2$ fixation. The non-isocyanate polyurethanes prepared via this method are being explored for their potential use in applications such as paints and adhesives.

The synthesis of cyclic carbonates has undergone incredible progress over the last decade [25]. Some researchers have developed cyclic carbonates via homogeneous metal catalysis [26–28] to solve challenges in the area. More recently, others have investigated metal-free approaches. Intrinsically, the organocatalytic activation of cyclic ether is less powerful than the metal-based conversions. However, in recent years, metal–organic frameworks (MOFs) have emerged as excellent candidates for heterogeneous catalysis $CO_2$ coupling reactions [29,30].

To date, a multitude of porous heterogeneous catalysts have been investigated in this reaction [31–37] due their desirable high surface area, tunable components, and pore sizes. In general, MOFs containing active Lewis acidic sites can promote the $CO_2$ fixation process, either with or without the use of a nucleophilic reagent [38]. The reaction of carbon dioxide with epoxides is one of the reactions where MOFs have been found to be efficient catalytic systems [29,30]. There have also been some investigations into the reaction of $CO_2$ with epoxides in the presence of MOF and an external co-catalyst [39]. The use of an external co-catalyst solves challenges related to the diffusion of reactants into the pores of the catalytic system. Nonetheless, a handful of MOFs catalysts have been reported for the successful $CO_2$ coupling reaction with bis-epoxides.

In this present work, several MOF catalysts have been investigated in order to improve the synthesis of bis-cyclic carbonates from bis-epoxides. MOFs are a class of crystalline polymer materials created from metals and organic linker compounds with a wide range of applications. MOFs are great for gas storage, separation, drug delivery, and catalysis, among other applications.

MOFs can be considered as an alternative approach to homogeneous catalysts because they effectively overcome the problem of the homogeneous catalysts, their synthesis is less complex, they are cheaper, and it is much easier to remove them [40]. Additional advantages over homogeneous catalysts include their high thermal, chemical, and mechanical stability, long life, renderability, extensive application potential, large pore size and porosity, high surface area, low density, and great versatility, to name just a few. To summarize, the main MOF characteristic is porosity, which allows organic–inorganic hybrid materials, such as zeolites, to reach record surface areas of up to 6000 $m^2/g$ and which also makes it possible for them to absorb vast volumes of gases such as $H_2$, $CH_4$, or $CO_2$. However, there are also some drawbacks: their selectivity can be variable, they have diffusion problems,

the reaction conditions are severe, their solubility is limited to particular conditions, and their mechanism is less affordable [41].

Their use as heterogeneous catalyst, particularly for the storage and separation of gases, is one of the most widely employed applications of MOFs. MOFs have numerous advantages over the zeolites, such as the possibility for direct incorporation of the catalytic metal centers. With MOFs, there is also the option to modify the environment of the catalytic centers, thanks to the functionalization of the organic ligands and the pore size, allowing to adjust the selectivity [42]. Metallic nodes play a very important role in the selectivity and show better stability compared to other catalytic systems.

MOFs can act as observer species where the catalytic reaction takes place or can be the active site that is capable of stabilizing the transition state and orienting the molecules. In some cases, the catalytic activity of MOFs can be attributed to the presence of unsaturated metal centers or catalytic species existing in the pores of the MOFs. Catalysis can take place also at the organic linker. Post-synthetic modifications can be carried out in order for the MOF to display catalytic activity at both the inorganic and organic sections [43].

MOFs also have been employed in other fields, such as in polymer research, where the creation of MOF and polymer nanocomposites can yield unprecedented new materials with pioneering features, which would not be achievable with only the individual components. Several MOFs have a great affinity for $CO_2$; both their porosity and structure tunability make them excellent prospective catalysts for the formation of bis-cyclic carbonates by $CO_2$ and bis-epoxides through coupling reactions.

In particular, the MOFs used in this work were M-MOF-74 and Al-PDA and Al-OH-fumarate. MOF-74 was selected due to its inherently large $CO_2$ adsorption capacity and selectivity, which is a result of the hexagonal and porous structure of MOF-74. Al-PDA and Al-OH-fumarate were selected because they can be characterized by their mechanical, thermal, and chemical stability, which implies a long catalytic life. Specifically, Al-PDA is characterized by having polydopamine (PDA), which could be used as an effective nucleation center for MOF deposition.

The metal catalytic center used in this work will act as a Lewis acid activating the epoxide; then, this favors nucleophilic halide attack, which results in the ring opening. Later, $CO_2$ can be inserted into the metal–oxygen bond, leading to a cyclic carbonate product. This mechanism can be seen in the figure below (Figure 1) [15,44–46].

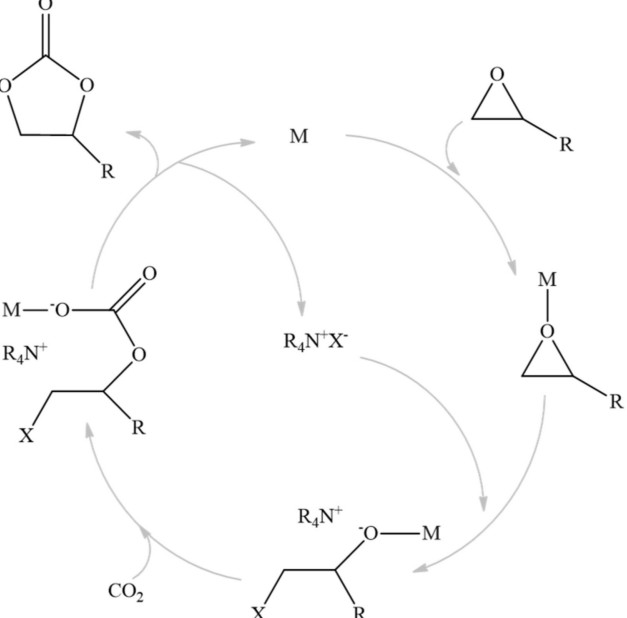

**Figure 1.** Mechanism of the reaction.

This work intends to show an alternative pathway of the coupling reaction of $CO_2$ with epoxides using heterogeneous catalysts easily available in the market, and with good thermal and chemical stability that allows them a long useful life. The coupling reaction carried out in this work is about diepoxides modification, as it is the most direct way to obtain polyurethanes.

## 2. Results and Discussion

### 2.1. Characterization of Catalysts

MOFs have not been previously reported as a typical solid catalyst for epoxide-involved reactions. As a result, it has been proposed as a catalyst for the coupling reaction of epoxides and $CO_2$ to test the catalytic activity. First, all catalysts have initially subjected to analysis to confirm the structure and adsorption capacity.

FTIR spectra are shown in Figure 2. The Al-OH-fumarate MOF exhibits the characteristic carboxyl functional group absorption peaks at 1700 $cm^{-1}$. The adsorption peaks at 1600 and 1500 $cm^{-1}$ are attributed to the asymmetric stretching vibration of –COO, whereas the symmetric stretching vibration appears at 1413 $cm^{-1}$. This implies that the -COO is coordinated to the aluminum center. Furthermore, in the range of 730 and 1000 $cm^{-1}$, the adsorption peaks could be attributed to the stretching vibration of CH, indicating the presence of the benzene ring structure connected to the catalyst. The vibration peak in the low wavenumber of 470 $cm^{-1}$ is due to the presence of Al-O bonds. On the other hand, at approximately 3400 $cm^{-1}$, there is a peak that corresponds to the -OH stretching vibration from free water or -OH groups present in the synthesized samples. For the Al-PDA FTIR spectra firstly, there is a peak at 3354 $cm^{-1}$ that is assigned to the bridging -OH vibration. The peak at 2929 $cm^{-1}$ could be attributed to the C-H stretching vibration. The bands at 1531 and 1380 $cm^{-1}$ are potentially the asymmetric and symmetric stretching vibrations of COO, respectively. PDA shows absorption peaks at 1500 $cm^{-1}$, which are a result of the aromatic C-C stretching vibrations. The FTIR spectrum of Co-MOF-74 shows a peak at 1544 $cm^{-1}$, which is attributed to the interaction between ligands and Co ions. Hence, the absorption peak at 808 $cm^{-1}$ will correspond to the Co-O stretching vibrations, indicating the existence of oxygen vacancies within the MOF. In the Mg-MOF-74 spectra, the weak absorption band observed at 1577 $cm^{-1}$ is due to the bending vibration of absorbed water, and the absorption band at 3207 $cm^{-1}$ is assigned to stretching of the O-H. The peak at 1421 $cm^{-1}$, confirms the presence of $NO_3$ counter ion in the structure. The band at 400 $cm^{-1}$ is a result of the of Mg-O bond. The Ni-MOF-74 FT-IR spectra shows the characteristic symmetric and asymmetric stretching vibrations of carboxylate groups at 1600 and 1400 $cm^{-1}$, respectively. The sample also exhibits characteristic peaks at 1192 and 890 $cm^{-1}$, which are regions attributed to C-H bending vibrations. In the case of GO/Ni-MOF-74 FTIR spectra, the same characteristic peaks of Ni-MOF-74 can be observed with the largest difference being that of the GO absorption peaks. There is a broad peak at 3260 $cm^{-1}$ corresponding to the O-H stretching of water molecules absorbed onto the GO. A sharp peak at 1664 $cm^{-1}$ was attributed to the C = O stretching vibration of carboxylic acid. The presence of an absorption peak at 1600 $cm^{-1}$ can be attributed to the aromatic C = C stretching vibration. The absorption peaks at 1100 $cm^{-1}$ and 1047 $cm^{-1}$ correspond to epoxy C-O stretching vibration and alkoxy C-O stretching vibration.

The porosity of Al-PDA catalyst was measured by nitrogen adsorption isotherms at $-77$ K. A type IV adsorption/desorption isotherm and a small hysteresis loop could be observed in Figure 3a. The isotherm indicated the presence of multi-layer adsorption on mesoporous materials. The adsorption increased sharply in the range of $p/p_0 = 0.5–0.95$, which was attributed to $N_2$ gas capillary condensation in the mesopores. The BET specific surface area obtained was 625 $m^2/g$; this means there is a large number of active sites and available surface area for reacting. A pore volume 0.24 $cm^3/g$ and an average pore diameter of 4.80 nm were also found. All of the results were similar to that reported in the literature [44]. In the case of Al-OH-fumarate, a type I adsorption–desorption isotherm was observed. This catalyst also shows a hysteresis loop in the partial pressure range of

0.5–1.0, as is observed in Figure 3a. This typical I isotherm revealed a microporous material: there is a sharp increase in the volume adsorbed at very low relative pressures followed by a nearly constant plateau up to $p/p_0 = 0.8$. The rise in the adsorbed volume at relative pressures close to 1 is indicative of the presence of mesopores and macropores between the particles. The BET specific surface area obtained is 667 $m^2/g$, which is quite lower than that reported in the literature [47]. A pore volume of 0.21 $cm^3/g$ and an average pore diameter of 3.09 nm were found, which shows the microporous characteristics of the material. Lastly, M-based MOF-74 catalysts all show a type I isotherm without a hysteresis loop (Figure 3a). They all have permanent microporosity with a BET specific surface areas as follows: Ni-, GO/Ni-, Co-, and Mg-MOF-74 of 527, 265, 31, and 830 $m^2/g$, respectively, which was similar to that reported in the literature [48], a pore volume of 0.23, 0.28, 0.03, and 0.40 $cm^3/g$, respectively, and an average pore diameter of 2.33, 4.15, 3.16, and 2.23 nm, correspondingly. The data are summarized in Table 1.

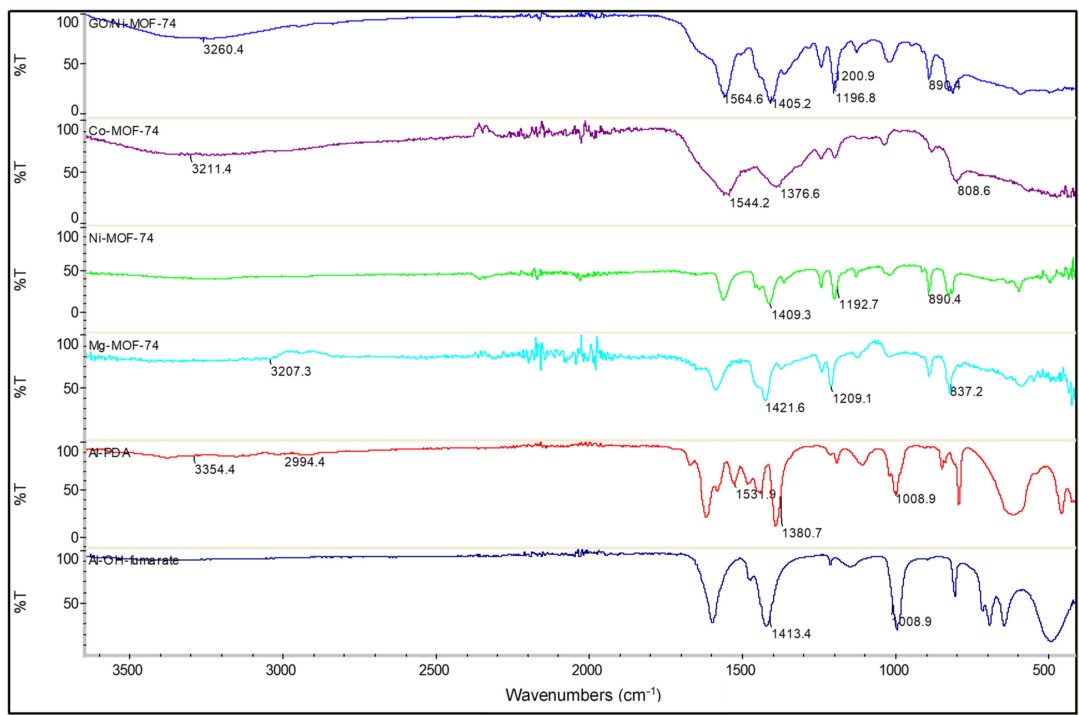

**Figure 2.** FTIR spectra of the MOFs. GO/Ni-MOF-74 (blue), Co-MOF-74 (**purple**), Ni-MOF-74 (**green**), Mg-MOF-74 (**light blue**), Al-PDA (**red**), and Al-OH-fumarate (**dark blue**).

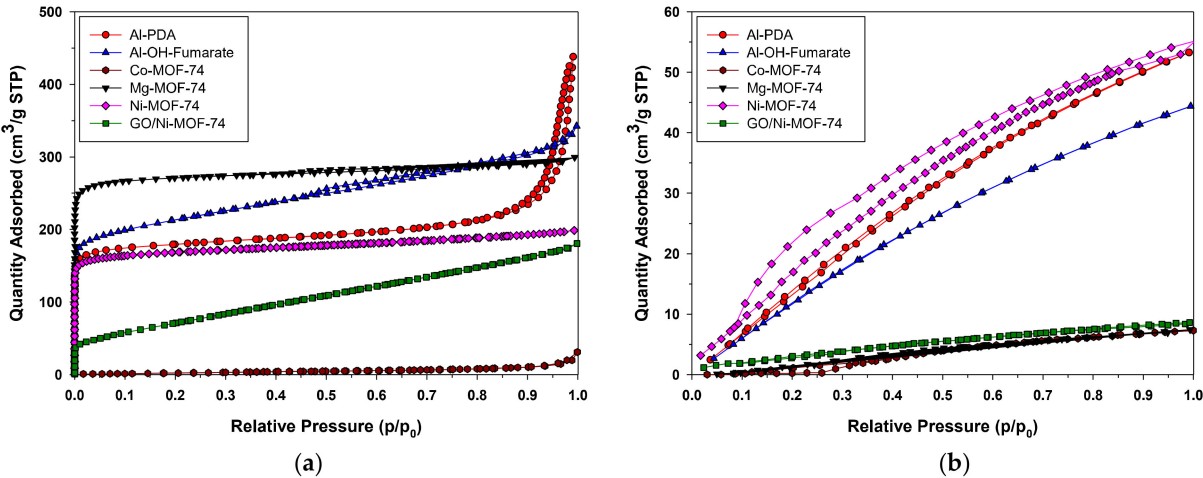

**Figure 3.** $N_2$ (**a**) and $CO_2$ (**b**) adsorption–desorption isotherms.

**Table 1.** Summary of results obtained from $N_2$ and $CO_2$ adsorption isotherms.

| Catalyst | BET $m^2/g$ | Pore Volume $cm^3/g$ | Pore Diameter nm | $CO_2$ Adsorbed Quantity * $cm^3/g$ |
|---|---|---|---|---|
| Al-PDA | 625 | 0.24 | 4.80 | 53.20 |
| Al-OH-fumarate | 667 | 0.21 | 3.09 | 44.30 |
| Ni-MOF-74 | 527 | 0.23 | 2.33 | 90.20 |
| GO/Ni-MOF-74 | 265 | 0.28 | 4.15 | 8.60 |
| Co-MOF-74 | 31 | 0.03 | 3.16 | 7.30 |
| Mg-MOF-74 | 830 | 0.40 | 2.23 | 7.44 |

* Conditions: 25 °C, 1 bar.

Carbon dioxide adsorption isotherms were recorded at 25 °C. Both the Al-PDA catalyst and Al-OH-fumarate MOF isotherms show a gradual increase in quantity adsorbed as the relative pressure rises, as shown in Figure 3b, which could be related to a mechanism of adsorption of $CO_2$ in the coverage of the surface, which is associated with a range of pore sizes two times the molecular dimension of the $CO_2$. In the case of Al-PDA, the maximum quantity of $CO_2$ adsorbed is 53.20 $cm^3/g$ STP, and for Al-OH-fumarate, the maximum quantity of $CO_2$ adsorbed is 44.30 $cm^3/g$ STP. Lastly, for M-based MOF-74, the $CO_2$ adsorption isotherm, as shown in Figure 3b, shows no hysteresis, meaning that the desorption section of the $CO_2$ isotherm is fully reversible. This suggests that M-based MOF-74 could be regenerated under mild conditions. The strong interaction of Ni-MOF-74 with $CO_2$ and relatively small pore volume of the structure result in a $CO_2$ adsorptive working capacity of 90.20 $cm^3/g$ while GO/Ni-, Co-, and Mg-MOF-74 have an adsorptive working capacity of 8.60, 7.30, and 7.44 $cm^3/g$, respectively.

The one-dimensional nanostructure and high surface area of the catalysts could allow them to be well dispersed in the monomers, allowing for improved contact between the active sites to the reactants. The quantity of $CO_2$ absorbed is shown in Table 1.

The scanning electron micrographs of MOF catalysts (FESEM images), taken at the same magnification in order to identify key structural features, are shown in Figure 4. The Al-PDA catalyst displays an irregular shape with a rough surface. This MOF was wrapped by a layer of film, which indicated that the polymeric layer of PDA was coated onto the surface. Elemental analysis was used to confirm this; the composition of the material showed that the percentage weights of C, N, O, and Al were approximately 58.54%, 9.5%, 28.02%, and 3.94%, respectively.

The image of Al-OH-fumarate shows a rough and porous structure. It is formed of irregular clusters where it is not possible to identify individual crystals. The wt % compositions of C, O, and Al were 56.38%, 36.96%, and 6.66%, respectively.

M-MOF-74 displays a prismatic and regular morphology starting from Co-MOF-74, which exhibits a rodlike form with a wt % composition of C, O, and Co around 65.02%, 26.76%, and 8.21% respectively. Continuing with Mg-MOF-74, a column-like morphology was observed with cauliflower shapes consisting of agglomerated needle crystals. The composition of the material reveals a wt % of C, O, and Mg approximately 69.9%, 27.41%, and 2.68% respectively. Finally, GO/Ni-MOF-74 and Ni-MOF-74 appear to be formed of sheets consisting of shorts hexagonal prisms with a uniform particle size. GO/Ni-MOF-74 is composed of 59.58 wt % of C, 27.44 wt % of O, and 12.99 wt % of Ni. However, the Ni-MOF-74 consists of 39.8 wt % of O and 60.2 wt % of Ni. All M-MOF-74 were found to have a uniform distribution of metal particles in the catalyst.

X-ray diffraction patterns (XRD) were used to confirm the phase structure of the MOFs used in this work. The XRD pattern were in accordance with those reported in the literature. From Figure 5a, the X-ray diffraction pattern of Al-PDA contains the expected significant peaks at 2θ = 8.2°, 15°, and 18.5°. The Al-OH-Fumarate XRD pattern shows the expected five distinct diffraction peaks at 2θ = 10.2°, 15.6°, 21.7°, 31.8°, and 42.7°, which are in good agreement with the reported standard peaks [49]. Regarding M-MOF-74, Co- and Mg-MOF-74 show an identical XRD pattern, exhibiting the isostructural nature of the materials as reported in the literature [50]. In addition, it can be seen in Figure 5b that

M-MOF-74 shows two distinct peaks at 2θ = 6.7° and 11.7°, with the remaining small peaks not so sharp and clear, which again is in accordance with reports in the literature [50,51].

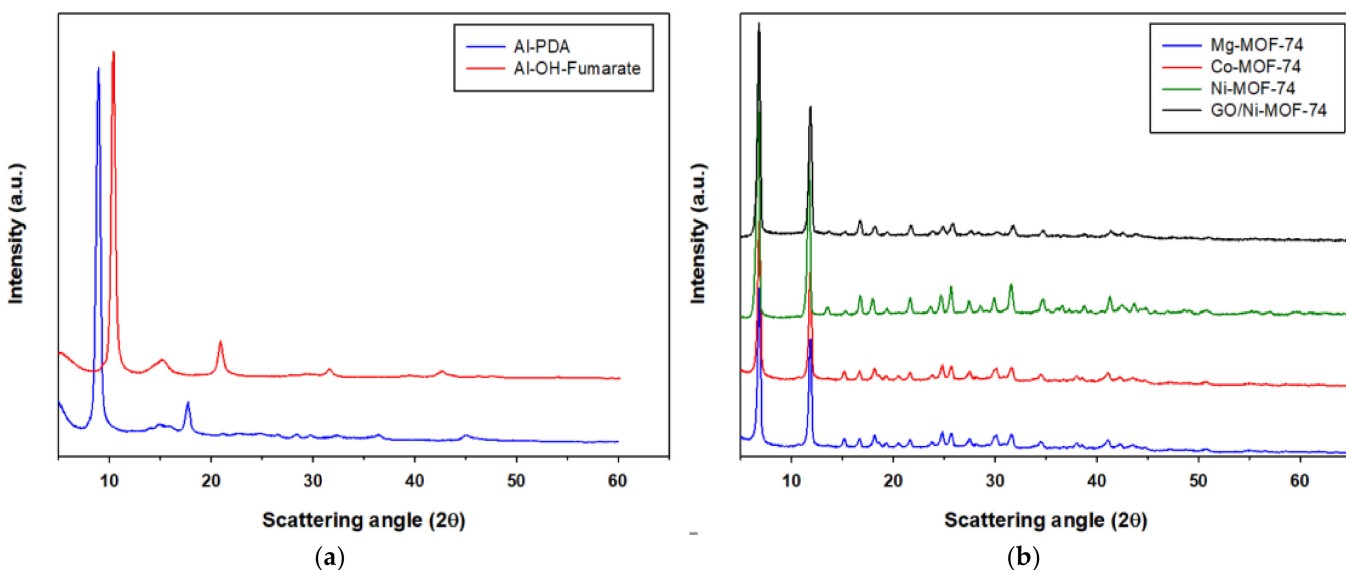

**Figure 4.** HRSEM images of (**a**) Al-PDA, (**b**) Al-OH-fumarate, (**c**) Mg-MOF-74, (**d**) Co-MOF-74, (**e**) GO/Ni-MOF-74, and (**f**) Ni-MOF-74.

**Figure 5.** XRD patterns of Al-PDA and Al-OH-fumarate (**a**) and M-MOF-74 (**b**).

## 2.2. Reaction

The $CO_2$ addition onto an epoxy group is a valuable experimental technique to prepare five membered cyclic carbonates. The carbonation of 1,3-butadiene diepoxide in the presence of a MOF catalyst yields BDC, as illustrated in Scheme 3. The reaction was carried out at approximately 120 °C and 30 bar $CO_2$ pressure, and vitally, it did not require either solvent addition or the removal of by-products.

In some cases, CTAB has been added to the reaction by researchers. CTAB is a quaternary ammonium cationic surfactant which acts as a template, thus avoiding catalytic agglomeration. CTAB is a micellar catalyst, so it can form micelles in the solution, which allows the product to grow. According to this behavior, it was decided that CTAB would be used as the co-catalyst [52,53].

Here, the coupling reaction yield was studied, using different catalysts and CTAB as co-catalyst under varying conditions. The primary purpose was to obtain the optimum green pathway with the higher yield. To date, 16 MOFs and their derivatives have been reported to catalyze this type of reaction using 11 different epoxides [54–56]. However, none of those MOFs have been probed with 1,3-butadiendiepoxide. Recently, Pander et al. [29] probed the coupling reaction with 1,3-butaiendiepoxide but using Zr-MOFs. In this work, six different MOFs have been used in the coupling reaction. The same molar ratio has been used in every reaction (catalyst: co-catalyst 1:1) at 120 °C and the reaction time were either 6 or 24 h.

As can be seen in Table 2, the reaction time played an important role in this reaction. The yield (NMR conversion) of the product decrease from 34% to 33% (entry 1 and 2) when the time was increased from 6 to 24 h. Due to this, every reaction was carried out with a reaction time of 6 h.

**Table 2.** Carbonation of 1,3-butadiendiepoxide.

| Entry | Catalyst | Catalyst wt % | Co-Catalyst %mol | T h | NMR Conversion [a] % | GC-MS Conversion % | TON | TOF |
|---|---|---|---|---|---|---|---|---|
| 1 | - | - | 0.01 | 6 | 34 | 27.2 | 352.50 | 58.75 |
| 2 | - | - | 0.01 | 24 | 33 | 26.4 | 327.19 | 13.63 |
| 3 | GO/Ni-MOF-74 | 0.36 | - | 6 | 1.8 | 1.3 | 17.94 | 2.99 |
| 4 | Ni-MOF-74 | 0.36 | - | 6 | 0 | 0 | 0 | 0 |
| 5 | Co-MOF-74 | 0.29 | - | 6 | 2 | 1.4 | 20.08 | 3.35 |
| 6 | Mg-MOF-74 | 0.29 | - | 6 | 0.6 | 0.4 | 5.75 | 0.96 |
| 7 | Al-OH-fumarate | 0.18 | - | 6 | 26.4 | 21.7 | 361.65 | 60.28 |
| 8 | Al-PDA | 0.24 | - | 6 | 11.3 | 6.9 | 106.59 | 4.44 |
| 9 | Co-MOF-74 | 0.29 | 0.01 | 6 | 65.3 | 46.1 | 648.04 | 108.01 |
| 10 | Al-OH-fumarate | 0.18 | 0.01 | 6 | 95.5 | 78.6 | 1304.51 | 217.42 |
| 11 | Al-PDA | 0.24 | 0.01 | 6 | 73.4 | 44.6 | 717.47 | 119.58 |

**General conditions:** Bis-epoxide (26 mmol), $CO_2$ (30 bar) PDA = polydopamine, GO = graphene oxide, T = 120 °C, Turn Over Number (TON), Turn Over Frequency (TOF). [a] NMR conversion of the bis-cyclic carbonate.

A portion of the reaction crude was brought under a column chromatography to purify it and it was used as a GC-MS pattern. Then, to determine the consequential conversion, every reaction crudes were analyzed by GC-MS. The product was also submitted for an NMR and FTIR analysis to confirm its identity.

As it can be seen in Table 2, the GC-MS and NMR conversion differs from each other, the first being approximately 20% lower. This is probably due to a combination of the NMR inaccuracies and the lack of purification of all the tested samples for GC-MS. The most interesting information is the relative effectiveness between catalysts, as well as the range of product conversion.

The experimental results also indicate that the conversion of BDC significantly improves when CTAB is added to the reaction. Hence, this carbonation represents an atom-efficient route to BDC.

The product obtained was submitted for an FTIR analysis to confirm the presence of all the expected functional groups. Comparing the FTIR spectra, as shown in Figure 6, the epoxy groups, which can be assigned to the transmission band at 909 cm$^{-1}$ in 1,3-butadiendiepoxide, disappears in BDC, while a new band at 1700–1800 cm$^{-1}$ appears, which is a result of carbonate carbonyl group. This proves that the epoxy groups have been converted into cyclic carbonates. As is shown in Figure 6, the produced bis-cyclic carbonate displays the strongest peak at 1700–1800 cm$^{-1}$, which has been assigned to the carbonyl group of BDC. The 1200 and 1100 cm$^{-1}$ peak likely corresponds to the C-O stretches. The stretching vibration of the C-H bond can be observed at 3000 cm$^{-1}$. Furthermore, the small but wide band observed at 3300 cm$^{-1}$ may be a result of the secondary hydroxyl groups generated during the urethane formation, or it could be assigned to small amounts of hydroxyl groups in their structures. Finally, those at 1654 and 1474 cm$^{-1}$ have been assigned as the vibration deformation of the CH$_2$ groups.

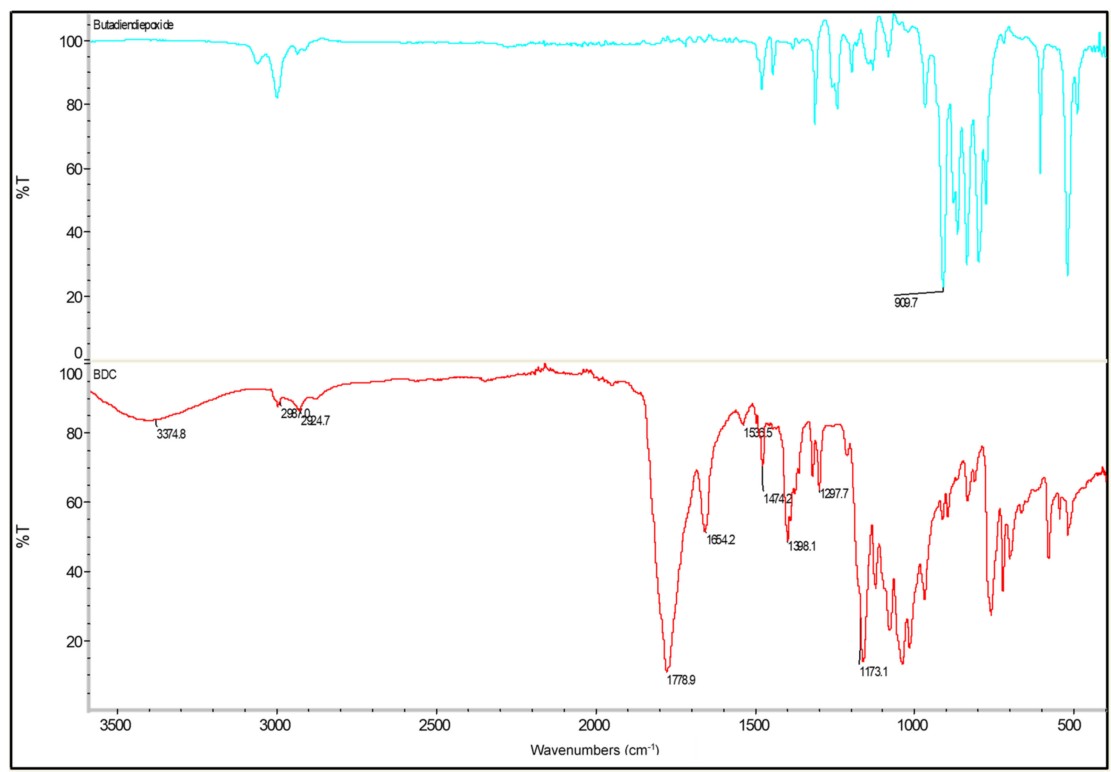

**Figure 6.** FTIR spectra of 1,3-butadiendiepoxide (**blue**) and BDC (**red**).

The cyclic carbonate formed from the epoxy group also underwent NMR analysis to verify the chemical structure of the molecule. Figure 7 shows that the chemical shifts of the methylene and methine protons of the epoxy group in 1,3-butadiendiepoxide at 2.68, 2.79, and 2.85 ppm disappeared after the coupling reaction, and new peaks at δ = 4.25–4.47 (m, 2 H, 1-CH$_2$), 4.60–4.70 (t, 8.8 Hz, 2 H, CH$_2$), 5.04–5.13 (m, 2 H, CH) ppm appeared. These results indicated the successful synthesis of BDC.

Hence, it has been confirmed by both NMR and FTIR spectroscopy that this route results in a good yield containing 50 wt % chemically fixed CO$_2$.

Likewise, a clear synergistic effect of the MOF and CTAB on the coupling reaction was noted. When solely the MOF catalyst was used, the BDC GC-MS conversion only reached 21.73%. When just CTAB was used, the BDC GC-MS conversion reached 27.21%. However, BDC GC-MS conversion of 78.61% was obtained with the binary catalyst of Al-OH-fumarate and CTAB. This result suggests that Al-OH- fumarate is the best catalyst for the bisepoxy-CO$_2$ coupling reaction, with superior results to those previously reported in the literature. Wei et al. [52] proposed a mechanism for this reaction, catalyzed by a

binary catalyst system of MOF and CTAB. The aluminum ions on the surface of the MOF act as a Lewis acid site, which activates the oxirane groups of the epoxides. Then, the bromine anion of CTAB adopts the role of a strong nucleophilic reagent and attacks the carbon atom of the activated oxirane groups, promoting the ring-opening reaction of the oxirane groups, which then results in a metal–oxyalkyl bond forming. After the insertion of $CO_2$ into the Al-O or Ti-O bond, a metal–carbonate species was obtained, leading to the five membered cyclic carbonates. The one-dimensional nanostructure and high surface area of the catalysts could allow them to be well dispersed in the monomers, allowing for improved contact between the active sites and reactants.

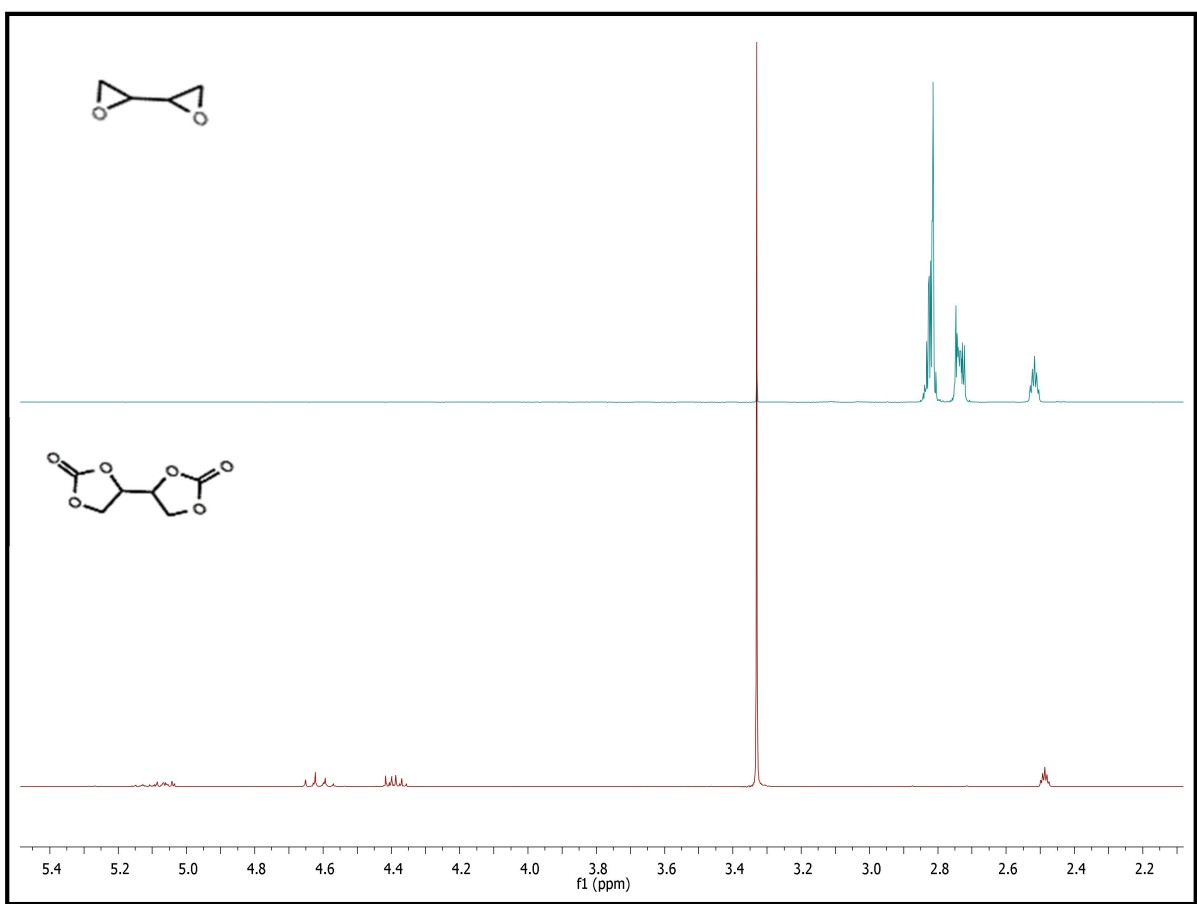

**Figure 7.** [1]H-NMR of 1,3-butadiendiepoxide (**blue**) and BDC (**red**) in DMSO-d6.

This study elucidates how the choice of the catalyst governs the yields of CC production.

A future approach could be synthesizing poly(hydroxyurethanes) via a polyaddition with primary alkyl amines such as HMDA (hexamethylenediamine). The cyclic carbonate–amine reaction that leads to the formation of urethane linkages proceeds slowly at lower temperatures. The nucleophilic attack by the amine on the CC ring is followed by the subsequent formation of the urethane linkages and secondary hydroxyl groups. It is possible to carry out polyaddition reactions of five membered cyclic carbonates and 1,6-hexamethylenediamine using CTAB as catalyst, at 60 °C over 5 h, resulting in five linear polyurethanes.

## 3. Materials and Experimental Section

### 3.1. Materials

1,3-Butadiene diepoxide (97%), dimethyl sulfoxide-d6 (DMSO-d$_6$), silica gel 60 (Munich, Germany,) and hexadecyltrimethyl-ammonium bromide (98%, CTAB) from Tianjin, China were purchased from Sigma-Aldrich and directly used. Carbon dioxide was pur-

chased from Linde (Valencia, Spain). All catalysts were obtained from MOF Technologies Company (Belfast, UK), and acetone was purchased from VWR (Barcelona, Spain) chemicals and n-Hexane was purchased from VWR (Paris, France). Finally, ethyl acetate was obtained from Scharlab (Barcelona, Spain).

### 3.2. Catalysts Characterization

*Nitrogen adsorption–desorption* isotherms at $-196\,^\circ$C were measured using a 3P micro 200 analyzer (3P Instruments, Odelzhausen, Germany)). Samples of approximately 0.2 g were previously in situ evacuated under high vacuum ($10^{-3}$ MPa) under different activation conditions. The micropore surface was calculated using the BET (Brunauer–Emmett–Teller) model [57]. The pore volume and diameter were estimated by non-local DFT (Density Function Theory) calculations, assuming a kernel model of $N_2$ at $-196\,^\circ$C on carbon [58] (assuming cylindrical pore geometry). The *adsorption isotherms* of $CO_2$ were carried out at room temperature, and gas pressures up to 800 mmHg were measured using the same equipment as the nitrogen isotherms. The desired temperature was achieved by using a dewar with a circulation jacket connected to a thermostatic bath. Approximately 0.1 g of adsorbed sample was used for the gas adsorption studies. *FTIR measurements* were obtained with a Nicolet 6700 FT-IR spectrometer from Thermo Scientific (Wisconsin, USA) using a diamond tip (Diamond 30,000–200 cm$^{-1}$) and ATR technique. *Scanning electron microscopy (SEM)* was carried out using a JEOL JSM-7001F (Tokyo, Japan). *XRD measurements* were performed on a 2D Phaser equipment (Bruker, Madrid, Spain) with Cu-Kα radiation working at 30 kV and 10 mA in order to confirm the crystal structures of MOFs.

### 3.3. Product Characterization

*NMR spectroscopy*: NMR spectroscopy measurements were recorded at a frequency of 300 MHz using the spectrometer Bruker AvanceIII 300 (Barcelona, Spain), at room temperature. DMSO-d$_6$ was used as deuterated solvent. All reported NMR values are given in parts per million (ppm). To confirm the product structure, it we used not only an NMR spectrometer but also an FTIR analyzer. *GC-MS conversions* were obtained using a GC Thermo Scientific Trace 1300 and MS Thermo Scientific TSQ 9000 from Illinois, USA. The GC-MS conditions used were a ZB-WAX (30 m × 0.25 mm × 0.25 μm) column, an injection volume of 2 μL, and helium as the carrier gas. The three analysis techniques allowed the molecular fingerprint of the sample to be identified.

### 3.4. General Procedure for the Carbonation of Bis-Epoxide

Cyclic carbonates were prepared via the coupling reaction of bis-epoxides in the presence of $CO_2$ in a stainless-steel autoclave (MiniClave, 190 mL, type Br-100, Berghof) equipped with a Teflon tube and an O-ring. The diepoxide was fed to the reactor along with the MOF catalyst (0.18–0.36 wt % of bis-epoxide). After that, the reactor was flushed three times with $CO_2$, 30 bar, and then, the autoclave was pressurized with 30 bar $CO_2$. The reactor was heated to 120 ºC, causing the pressure to rise. When the desired reaction temperature was reached, the pressure was found to be 40–45 bar; however, it stayed constant during the course of the reaction. The preferred stirring speed was 600 rpm. After either 6 or 24 h, the reaction was stopped and cooled down to room temperature by introducing the reactor into an ice bath. At room temperature, the reactor was depressurized slowly to allow any dissolved $CO_2$ to leave the carbonated product. The conversion of diepoxide to cyclic carbonate was confirmed by NMR and FTIR analysis. The reaction pathway is shown in Scheme 3.

**Scheme 3.** Synthesis of BDC.

### 3.5. Syntesis of Butadiene Dicarbonate (BDC)

Initially, 2 mL of 1,3-butadiene diepoxide was fed to the reactor with the MOF catalyst (molar ratio 1000:1). After that, the reactor was flushed three times with 30 bar $CO_2$, and then, the autoclave was pressurized with 30 bar $CO_2$. The carbonation took place at 30 bar and 120 °C over 6 h. The conversion of the diepoxide to cyclic carbonate was confirmed by NMR and FTIR analysis.

## 4. Conclusions

The carbonation of 1,3-butadiendiepoxide using an MOF as a catalyst results in high-purity BDC as a versatile and highly reactive dicyclic carbonate intermediate, enabling the production of non-isocyanate poly(hydroxyurethanes). Neither solvent nor by-product removal is necessary when BDC is produced via the coupling reaction of 1,3-butadien diepoxide with $CO_2$. The obtained BDC could be used to prepare polyurethanes with -OH groups via the polyaddition with HMDA.

The reaction time did not play an important role in the coupling reaction. The yield (GC-MS conversion) of the product decreased slightly from 27.21% to 26.40% when the reaction time was extended from 6 to 24 h.

Six different MOF catalysts were tested, and the results indicated that Al -OH-fumarate is the best catalyst, achieving a 21.73% GC-MS conversion. It was shown that with CTAB as a co-catalyst, the GC-MS conversion increased up to 78%. Thus, it can be concluded that there is a strong synergistic effect between the MOF and CTAB.

CTAB was chosen because is a commonly used co-catalyst in the coupling reaction of epoxides with $CO_2$ [52,53]. However, a future approach would be studying the importance of the co-catalyst choice in a binary catalysis with Al-OH-fumarate in order to optimize the reaction.

In addition, this reaction allows the final product to contain 50 wt % of fixed $CO_2$. This is a clear contribution to reducing greenhouse gases. Hence, it is clear that this reaction pathway is an ecofriendly way to produce bis-cyclic carbonates as a high value chemical building block.

**Author Contributions:** A.B. has contributed to the conceptualization, methodology, investigation (experimental design and reactions performance), review and editing, as well as funding acquisition. E.A. has focused her work on the investigation and writing—original draft preparation, while E.G. has mainly contributed to the methodology, formal analysis (HRSEM, adsorption isotherms), and review and editing of the initial draft manuscript. All authors have read and agreed to the published version of the manuscript.

**Funding:** This research was funded by IVACE, grant number IMIDEC/2019/5.

**Conflicts of Interest:** The authors declare no conflict of interest.

## Abbreviations

| | |
|---|---|
| BDC | [4,4′-bi(1,3-dioxolane)]-2,2′-dione |
| BET | Brunauer–Emmett–Teller |
| CC | Cyclic-carbonate |
| CTAB | cetyltrimethyl-ammonium bromide |
| DFT | Density Function Theory |
| DMSO-d6 | dimethyl sulfoxide |
| FTIR | Fourier transform infrared |
| GC-MS | gas chromatography mass spectrometry |
| HMDA | hexamethylendiamine |
| MOF | metal–organic framework |
| NIPHU | non-isocyanate polyhidroxyurethane |
| NMR | nuclear magnetic resonance |
| PHU | polyhydroxyurethane |
| PU | polyurethane |

| SEM | scanning electron microscopy |
|-----|------------------------------|
| TGA | thermogravimetric analysis |
| TLC | thin layer chromatography |
| TON | turn over number |
| TOF | turn over frequency |
| XRD | X-ray diffraction |
| 5CC | five-membered cyclic carbonates |

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
