# Peer review of "Highly Efficient MOF Catalyst Systems for CO2 Conversion to Bis-Cyclic Carbonates as Building Blocks for NIPHUs (Non-Isocyanate Polyhydroxyurethanes) Synthesis"

_catalysts, doi:10.3390/catal11050628_

Round 1
Reviewer 1 Report
Below are some questions/suggestions I have for the authors that should be addressed before accepting this manuscript.
- The work lacks the novelty of chemical synthesis. Similar type of work was published recently. There is no explanation of significant novelty of the work as compared with their previous work. This should be highlighted and compared with their previous work. What are the new findings in this work and how it differs from their previous work?
- In the introduction I’m missing reference and discussion of major work on cycloaddition of CO2 and epoxides, no reference to this substantial contribution to this area is given but furthermore, that research should be discussed in these context as it is highly relevant, for example: Inorg. Chem. 2018, 57, 2584–2593; Chem. – Eur. 2020, 26, 13686–13697; ACS Catal. 2017, 7, 3532–3539 but many others are available.
- Figure 1, Figure 2 and Figure 3 are schemes not figures.
- Chemical structures drawings are of poor quality (different fonts, in some places the atoms are not connected).
- Figure 5. FT-IR spectra are poor quality, especially Mg-MOF-74 and Co-MOF-74.
- There is lack of superscript for example at page 3 line17:, page 6, line 228, page 10 line 410.
- Table 1. BET values are given with too many decimal places.
- Please add necessary uncertainty of measurement (Table 1).
- Figure 9. FT-IR green spectrum, there is a negative peak, this need explanation.
- Residual solvent peak for DMSO-d6 on 1H NMR is 2.50 ppm, on Figure 10 there is a peak at 3.5 ppm, additionally, the blue spectrum is cut off and its scale is shifted.
- The yield and purity of the products obtained should be confirmed with GC-MS, which is today a gold standard in catalysis.
- The graphics in general are of poor quality.
- No information on which technique was used for IR measurements: KBr pellets, nujol suspension, ATR, DRIFT etc.
- Lack of PXRD study confirming the presence of the correct phase of MOFs.
- No comparison of the obtained results with the literature. How the systems presented are better than those already published? Some relevant references about CO2 cycloaddition reaction with 1,3-butadiendiepoxide on MOFs should be cited, such as: Front. Energy Res. 2015, 2, No. 63, ACS Catal. 2015, 5, 6748−6752, Nat. Commun. 2015, 6, No. 5933, ACS Appl. Mater. Interfaces 2021, 13, 8344–8352.
- The authors used CTAB as co-catalyst, what is activity with different co-catalyst?
- Table 2 yield value has too decimal places.
Author Response
1. The work lacks the novelty of chemical synthesis. Similar type of work was published recently. There is no explanation of significant novelty of the work as compared with their previous work. This should be highlighted and compared with their previous work. What are the new findings in this work and how it differs from their previous work?
We thank the reviewer for this comment asking us about more clarifications regarding the novelty of the work compared with the most recent literature. Several incremental steps are included in this work with respect to the literature consulted. In this way, it should be noted that we have opted for the family of dicyclic carbonates that have not been too explored in other works. Its importance lies in the fact that it is a very interesting building block for the subsequent direct synthesis of NIPUS (non-isocyanate polyurethanes) and HNIPUS (hybrid non-isocyanate polyurethanes). Moreover, this work is based on heterogenous catalysis, and a large part of the studies consulted refer to homogeneous catalysis (one of the authors of the recommended ACS Catal. 2017, 7, 3532–3539 has pointed us this issue). Studied catalyst systems are different MOFs structures, where Al-PDA and Al(OH)-fumarate, for example, are not deeply studied in the synthesis of cyclic carbonates, less in the case of dyciclic structures. The selection of these catalysts is motivated by their recognized thermal and chemical stability. In comparison, CPO (MOF-74) structures, reported in different articles as catalysts, have a lack of chemical and crystalline stability due its strong interaction with the water/humidity. Which is a plus point for the regeneration of the catalyst itself in future industrial applications. A paragrapfh in has been added.
2. In the introduction I’m missing reference and discussion of major work on cycloaddition of CO2 and epoxides, no reference to this substantial contribution to this area is given but furthermore, that research should be discussed in these context as it is highly relevant, for example: Inorg. Chem. 2018, 57, 2584–2593; Chem. – Eur. 2020, 26, 13686–13697; ACS Catal. 2017, 7, 3532–3539 but many others are available.
Thanks to the reviewer for highlighting this deficiency, and the literature recommendations. Deeper and preciser discussion about cycloaddition and epoxides conversion has been added in the introduction section. New bibliography and relevant studies have been included.
3. Figure 1, Figure 2 and Figure 3 are schemes not figures.
As the reviewer has pointed us, we have modified the nomenclature for figure 1, figure 2 and figure 3.
4. Chemical structures drawings are of poor quality (different fonts, in some places the atoms are not connected).
All the chemical structures have been redrawn using our version of chemdraw©. Type of fonts and size have been unified. In general the quality of the chemical structures and reactions have been improved.
5. Figure 5. FT-IR spectra are poor quality, especially Mg-MOF-74 and Co-MOF-74.
Quality of this figure has been improved. The font sizes of the spectra legend have been magnified for better reading quality.
6. There is lack of superscript for example at page 3 line17:, page 6, line 228, page 10 line 410.
These mistakes have been conveniently corrected. Thanks to the reviewer for the warning.
7. Table 1. BET values are given with too many decimal places.
8. Please add necessary uncertainty of measurement (Table 1).
Decimal places and uncertainty of the BET measurement has been corrected. In this case, confidence ratio of the BET polypoint fitting has been included in the table 1.
9. Figure 9. FT-IR green spectrum, there is a negative peak, this need explanation.
Negative peak is a mistaken. Sorry. We repeated the measurement and we added a baseline (we were not interested in a quantitative determination, but a identification of the species).
10. Residual solvent peak for DMSO-d6 on 1H NMR is 2.50 ppm, on Figure 10 there is a peak at 3.5 ppm, additionally, the blue spectrum is cut off and its scale is shifted.
Thanks for the advises related to the NMR spectras. On the one hand, regarding the peak at 3.5 ppm, it corresponds to the water signal of DMSO-d6 at 3.3 ppm, namely, the quantity of water dissolved in DMSO. On the other hand, the blue spectrum scale has been amended because, as the reviewer mentioned, it was not correctly represented.
11. The yield and purity of the products obtained should be confirmed with GC-MS, which is today a gold standard in catalysis.
We thanks the reviewer for pointing out this important issue. During this period we have isolated the product through a chromatographic column to use them as standards for GC-MS Measurements. Moreover, these products have been identified by FTIR and 1H NMR. Then, all the samples have been analyzed by GC-MS. The results are shown in the table 2 in the page 10. Some divergences with NMR have been found, but they are perfectly correlative. Both results, reinforce the conclusions regarding the effectiveness of the catalysts.
12. The graphics in general are of poor quality.
Thanks to the reviewer for underlighthing this issue. We have tried to improve the quality of all graphics, and pictures. For example, figure 4 has been completely redrawn to increase resolution and quality.
13. No information on which technique was used for IR measurements: KBr pellets, nujol suspension, ATR, DRIFT etc.
More details about FTIR measurements have been added to the experimental section. In detail, all the measurements were performed with ATR (attenuated total reflectance)-FTIR sampling technique. These modifications have been included in the final version of the article.
14. Lack of PXRD study confirming the presence of the correct phase of MOFs.
We agree with the reviewer on the lack of XRD measurements to confirm the correct cristalline structure. For this purpose the x-Ray powder diffraction analysis were performed on a 2D Phaser equipment with Cu-Kα radiation working at 30 kV and 10 mA in order to confir all cristal structures. The corresponding X-ray diffraction patterns have been included in experimental and discussion sections, as well as the analysis of the results.
15. No comparison of the obtained results with the literature. How the systems presented are better than those already published? Some relevant references about CO2 cycloaddition reaction with 1,3-butadiendiepoxide on MOFs should be cited, such as: Front. Energy Res. 2015, 2, No. 63, ACS Catal. 2015, 5, 6748−6752, Nat. Commun. 2015, 6, No. 5933, ACS Appl. Mater. Interfaces 2021, 13, 8344–8352.
My thanks to the reviewer because the recommended references are very helpful. One important advantage of the selected catalysts respect to the used in the reference from ACS Appl. Mater. Interfaces is the availability and stability of the catalysts. Moreover, our results cannot be compared with those of that article, since it does not indicate anything about the methodology for measuring yields, conversion and selectivity. Neither NMR nor GC-MS or FTIR is specified. MOF catalysts used in our work have the advantage that they can be currently manufactured at big scales. This confirms the capacity to reach higher TRLs in a short period of time, as well as the high stability facilitates handling, storage and regeneration procedures. Really this is our main objective.
16. The authors used CTAB as co-catalyst, what is activity with different co-catalyst?
Thanks to the reviewer for this interesting question. When we began the study, our main objective was to improve the yield of CO2 conversion using new generation of MOF structures. We had experience about these MOFs from previous works related to CO2 capture. In our mind, it want to unify capture and conversion in only one-step, avoiding the necessity of a regeneration of sorbents, or the high energy requirements for desorption process. Therefore these MOFs were selected for this reason. The non-expensive co-catalyst was not the objective of the work, but we knew this kind of anionic surfactants increase catalyst activity. We selected CTAB because its effectiveness it is well reported. We have included this bibliographic evidences in the experimental section.
During the study we have detected that the influence and synergic effect of CTAB co-catalyst with the MOF structures is huge. I agree with the reviewer that we would need to continue the work testing other similar co-catalyst to acquire more evidences and knowledge about this interaction. Future work will be focus on this issue.
17. Table 2 yield value has too decimal places.
Decimal places of every table have been corrected.

Reviewer 2 Report
The article describes MOF catalyst systems for CO2 conversion.
The manuscript is well-structured, although some concerns must be addressed:
- Introduction: Due to the fact that figure 3 presents a compounds that is claimed to be synthesized in this study, this figure should be moved in the experimental part.
- Please improve the resolution of figure 4.
- All figures and tables must be included after their first mention in the text.
- Figure 5: image and caption should be on the same page.
- Figure 5: please increase the font for the samples names.
- Figure 6: why in this image, the N2 desorption is not presented?
- Figure 6: please use superscript and subscript for the measurements units (cm3/g; P/P0).
- Table 2: please use the same number of decimals for all values.
- Figure 10: image and caption should be on the same page.
- “Experimental” section: please provide company, city, country for all reagents and instruments.
- The English language must be revised.
Author Response
- Introduction: Due to the fact that figure 3 presents a compounds that is claimed to be synthesized in this study, this figure should be moved in the experimental part.
Figure 3 has been moved to the experimental section to better coherence and clarity with the description of the synthesis process.
- Please improve the resolution of figure 4.
In general, all the chemical structures and specifically figure 4 have been redrawn using our version of ChemDraw©.
- All figures and tables must be included after their first mention in the text.
Some mistakes related to the location of figures and tables have been detected and corrected.
- Figure 5: image and caption should be on the same page.
The problems with the document structure have been solved. The location of all images and captions have been checked.
- Figure 5: please increase the font for the samples names.
Figure quality has been improved, increasing the size of the Font for the sample names. Axis reading quality has been also improved.
- Figure 6: why in this image, the N2 desorption is not presented?
We removed N2 and CO2 desorption processes to better clarity of the images. We have added these information in the same images trying to improve the clarity and readibility of the curves. For this purpose, Origin© software has been employed.
- Figure 6: please use superscript and subscript for the measurements units (cm3/g; P/P0).
This kind of mistakes have been solved. We have cheked the rest of the document to assure that superscripts and subscrips are correctly represented.
- Table 2: please use the same number of decimals for all values.
In general we have checked all the tables and results, modifying the significant number of decimals when it has been required.
- Figure 10: image and caption should be on the same page.
As was happening with figure 5, the placement of images and subsequent captions have been reviewed and corrected when necessary.
- “Experimental” section: please provide company, city, country for all reagents and instruments.
Whole information about reagents and instruments used in the work has been added. This includes company and city, as required.
- The English language must be revised.
The quality of English has been reviewed throughout the document. A native colleague has been in charge of this revision task.

Round 2
Reviewer 1 Report
The authors improved their manuscript significantly, it is now suitable for publication and I recommend publishing this manuscript.